# Monosomy 18p with Unbalanced Translocation Between 13 and 18 Chromosomes: First Reported Case in Serbia

**DOI:** 10.3390/diagnostics15030358

**Published:** 2025-02-04

**Authors:** Bojana Marković, Marina Gazdić Janković, Zoran Igrutinović, Raša Medović, Nevena Stojadinović, Biljana Ljujić

**Affiliations:** 1Pediatric Clinic, University Clinical Centre Kragujevac, Zmaj Jovina 30, 34000 Kragujevac, Serbia; bojana.kovacevic96@gmail.com (B.M.); igzor@medf.kg.ac.rs (Z.I.); rasamedovic@gmail.com (R.M.); niblackpearl@gmail.com (N.S.); 2Department of Pediatrics, Faculty of Medical Science, University of Kragujevac, Svetozara Markovica 69, 34000 Kragujevac, Serbia; 3Department of Genetics, Faculty of Medical Sciences, University of Kragujevac, Svetozara Markovica 69, 34000 Kragujevac, Serbia; bljujic74@gmail.com; 4Department of Communication Skills, Ethics and Psychology, Faculty of Medical Science, University of Kragujevac, Svetozara Markovica 69, 34000 Kragujevac, Serbia

**Keywords:** 18p deletion syndrome, 18p monosomy, unbalanced translocation, hypotonia, hypothyroidism, astigmatism

## Abstract

**Background**: Monosomy 18p is a chromosomal disorder resulting from the deletion of the short arm of chromosome 18. While a lot of cases result from the partial deletion of 18p, only a few reported cases are caused by the deletion of the whole short arm of chromosome 18 due to unbalanced translocations occurring between chromosomes 13 and 18 (13;18). 18p- monosomy presents with a variety of clinical manifestations, including facial dysmorphism, intellectual disability, and short stature, among others. **Case presentation**: Here, we report a case of a one-year-old girl with 18p- monosomy resulting from an unbalanced translocation between chromosomes 13 and 18 (45, XX, t(13;18) (q12:p11.2)). Our patient had facial dysmorphism and stunted growth. Additionally, she had hypotonia and required thyroxine supplementation from a young age. To our knowledge, this is the first case of astigmatism in a patient with this deletion and an unbalanced translocation between chromosomes 13 and 18. **Conclusions**: The present case demonstrates the phenotypic spectrum of a rare variant of monosomy 18 caused by an unbalanced whole-arm translocation between chromosomes 13 and 18. Our study emphasizes the significance of cytogenetic testing to diagnose this disease, which has been described only five times in the literature.

## 1. Introduction

A chromosomal disorder resulting from the deletion of all or part of the short arm of chromosome 18 is known as 18p deletion syndrome [1]. The majority of cases occur due to de novo deletions, although cases of direct parent-to-child transmission have been reported [2,3]. While most cases result from the terminal deletion of 18p, 16% of the cases that have been reported were as a result of an unbalanced whole-arm translocation resulting in monosomy 18p [2].

The clinical manifestation differs among individuals with different chromosome breakpoints, so patients with monosomy 18p can develop any symptom of 18p deletion syndrome. The most common features of the syndrome are intellectual disability, speech delay, and short stature [4,5,6]. Patients show facial dysmorphism, including ptosis, round face, microcephaly, a flat and broad nasal bridge, protruding ears, horizontal palpebral fissures, epicanthal folds, and strabismus. Many patients with this condition have refractive errors such as myopia and hyperopia, as well as dental and various skeletal deformities [4,7]. Endocrine abnormalities are also found in these patients, of which growth hormone deficiency is prevalent, isolated, or as a part of hypopituitarism, so growth factor treatment is justified [3,7,8]. Hypotonia is quite common in these patients [5,7]. 18p deletion may result in severe brain malformations such as holoprosencephaly in some cases [9,10]. The most common complications in the neonatal period in patients with complete 18p deletion include jaundice, respiratory difficulties, and feeding problems [6]. Patients may present with dystonia in young adulthood [11,12,13]. Although cardiac malformations are uncommon, some patents may have septal defects, Tetralogy of Fallot, or situs abnormalities [14,15]. Patients with 18p deletion may also develop autoimmune diseases such as Graves’ disease, rheumatoid arthritis, lupus, or psoriasis [7,16,17,18,19]. Additionally, a reduced serum level of immunoglobulins, most notably IgA, results in immune dysfunction and may contribute to chronic otitis media [3,7]. A cytogenetic analysis is necessary to make a definite diagnosis of 18p monosomy. Considering the variety of clinical manifestations in these patients, a multidisciplinary approach is needed to provide them with adequate medical care.

Here, we describe the clinical features and diagnostic workup of a patient with monosomy 18p due to unbalanced translocation between 13 and 18 chromosomes. To the best of our knowledge, a total of five patients have been reported so far, and this is the first case of 18p- syndrome in a Serbian patient.

## 2. Case Description

### 2.1. Signs and Symptoms

An 11-month-old infant was admitted to the Pediatric Clinic’s hematology department because of refractory anemia. The patient was the first child of healthy parents. She was born after a normal full-term pregnancy and delivery, and was small for her gestational age. The birth weight was 2890 g (p3) and length was 47 cm (<p3). During pregnancy, the mother had a urinary infection and used antibiotics. Because of neonatal jaundice, the girl was treated with phototherapy.

At 11 months of age, her weight was 7 kg, height was 67.3 cm, and head circumference was 43 cm, which is 2SDs (standard deviations) under the average. She showed generalized stunted growth and developmental delay. A head examination showed light blonde hair, light complexion, broad forehead, light eyebrows, a depressed and broad nasal bridge with light hyperpigmentation on the nasal tip, upslanting palpebral fissures, epicanthic folds, a deep and long philtrum, an almond-shaped palpebral fissure, a thin upper lip, and a high-arched palate. She had two lower teeth and dentition was delayed.

During a neurological examination, the patient had central hypotonia and assumed frog-leg posture lying down. The girl sat independently, bent forward. She could crawl and stand, but not walk. She had difficulties with swallowing and chewing. An EEG during spontaneous sleeping showed nonspecific bilateral fronto-central abnormalities.

The hematologists were looking for a cause of anemia. The girl came to hospital with a hemoglobin value of 102 g/L and erythrocyte count of 3.64 × 1012/L. The blood tests showed low levels of iron (5.1 μmol/L) with a low total iron-binding capacity (43.4 μmol/L) and low transferrin saturation (11.8%). Hemoglobin electrophoresis revealed no abnormalities for her age. Because of the feeding difficulties and development delay, a gastroenterologist was consulted. We excluded celiac disease by detecting no anti-tissue transglutaminase antibodies in the blood and cystic fibrosis by detecting normal levels of chloride in the sweat. An analysis of her feces showed no blood nor parasites in the stool. Allergy tests for nutritive allergens were negative.

The immunological tests presented a low level of serum IgG antibodies for the patient’s age and normal levels of serum IgA and IgM antibodies (IgG = 2.19 g/L, IgA = 0.13 g/L, IgM = 0.86 g/L). The population of mononuclear cells was analyzed by immunophenotyping of the peripheral blood. Mature B lymphocytes comprised 27% of all the mononuclear cells, while T-lymphocytes comprised 52% in a CD4:CD8 ratio = 3:1. NK lymphocytes comprised 10% of the analyzed cells. The absolute number of isotype-switched memory B lymphocytes was marginally reduced compared to the reference range according to age (Table 1).

Her hormonal status showed low free thyroxine levels (10.99 pmol/L), while the levels of thyroid-stimulating hormone were normal. The level of insulin-like growth factor was 15.0 ng/mL. The cortisol values were normal. An ophthalmological examination revealed astigmatism in both eyes.

A psychologist evaluated the patient when she was 1 year old based on information provided by the patient’s mother using the Brunet–Lézine Scale. The results of testing were under the expected norms for her age, with IR = 87, suggesting that the child’s psychomotor development was delayed by about 48 days. The developmental delay was most noticeable in the domain of gross and fine motor skills.

The parents gave written consent for the publication of this case report.

### 2.2. Genotype–Phenotype Correlation

The genotype–phenotype correlation was limited and included feeding difficulties and poor growth, minor facial dysmorphic features, global developmental delay with hypotonia, and speech delay.

### 2.3. Molecular Cytogenetics

A standard karyotyping analysis performed on peripheral blood lymphocytes using G-banding techniques revealed a female karyotype with an unbalanced translocation between the long arm of one chromosome 13 and the long arm of one chromosome 18, resulting in a karyotype of 45XX, t(13;18) (q12:p11.2).

Molecular karyotyping was carried out using the SurePrint G3 Human CGH array kit 8X60K following the manufacturer’s instructions (Agilent Technologies, Santa Calara, CA, USA, UCSC hg19, NCBI Build 37, February 2009) in order to identify the size and location of genetic deletions/gains which often accompany chromosomal abnormalities. The results were analyzed with the CytoGenomic 5.1 (Agilent) program.

The parents’ cytogenetic assessment revealed that they had normal karyotypes, indicating that a de novo translocation between the long arms of chromosome 13 and 18 had resulted in 18p- syndrome in this case.

The standard karyotyping analysis detected a female karyotype with an unbalanced translocation between chromosomes 13 and 18: 45XX, t(13;18) (q12:p11.2).

The molecular karyotyping of the patient’s genomic DNA taken from peripheral blood revealed a deletion of the short arm from the 18p11.32 to p11.21 region (14.79 Mb) affecting 191 genes, of which 65 were protein-coding genes (Figure 1). Despite the large size of the deletion, its precise size and location had not been detected previously by karyotyping. The results were consistent with chromosome 18p deletion syndrome.

### 2.4. Treatment

There is no causative treatment for 18p monosomy. The treatment is directed towards alleviating specific symptoms and depends on clinical manifestations. Early developmental stimulation and treatment from speech therapists and oligophrenologists are very important because of developmental delays. The physiatrist’s stimulation treatments must be intensive. This patient should be followed by an ophthalmologist because of the astigmatism. Regular check-ups with an endocrinologist and further monitoring of growth and pituitary function are important because of the patient’s hypothyroidism. Lifelong thyroxine therapy is necessary and depends on the blood hormone levels. A neurologist should follow the patient’s further development with periodical EEG monitoring because of previous nonspecific changes and potential seizures, which could be a manifestation of this syndrome. Our patient needs regular hematologist and immunologist check-ups because of her anemia, low levels of IgG immunoglobulins, and reduced absolute number of memory B lymphocytes. Her iron deficiency should be corrected with iron supplementation.

## 3. Discussion

The incidence of monosomy 18p is estimated to be about 1:50,000 live-born infants, with a female to male ratio of 3/2 [5]. To the best of our knowledge, there are only five previously published cases of unbalanced translocations occurring between chromosomes 13 and 18 (13;18) associated with 18p- syndrome [20,21,22,23,24], and there are none in the Serbian population. The first known case in the world was described in a 27-year-old male in 1979 [20]. Herein, we report a rare case of a female infant diagnosed with 18p- syndrome as a consequence of a de novo unbalanced whole-arm translocation between chromosomes 13 and 18 (Appendix A). Three previously described patients also had a de novo translocation, and for two of them, there were no data about inheritance. At the time of writing this article, the current patient is the youngest patient diagnosed with this condition, and the second female [20,21,22,23,24].

Our patient had early nonspecific signs of this disorder in the neonatal period. She had neonatal jaundice and generalized hypotonia, which persisted until she was 11 months old and admitted to our hospital [5,7]. While these signs could be the first manifestation of monosomy 18p in the neonatal period, there was no evidence of this feature in five other patients described with this deletion and unbalanced translocation between chromosomes 13 and 18 [5,7,20,21,22,23,24]. On physical examination, we noticed minor facial dysmorphism, stunted growth, and developmental delay [4,5,6]. Regarding the fact that she was born small for her gestational age, her weight and height were still 2SDs under the average [5,6]. Low levels of insulin-like factor 1 (IGF-1) do not have strong significance at her age. Two previously described patients who had this syndrome had low birth weights [21,22], while all of them had short stature [20,21,22,23,24].

One of the main features of monosomy 18p is facial dysmorphism. In accordance with all the reported patients with this syndrome, together with the unbalanced translocation between chromosome 13 and 18, our patient had some dysmorphic characteristics [4,5,20,21,22,23,24]. There is no pattern which is always presented, but there are some manifestations that are more common. Almost every patient reported with this condition had a different head shape. All of them presented with eye disorders. The most common issue was strabismus, followed by ptosis, epicanthic folds, hypertelorism, and others [20,21,22,23,24]. Ear malformations were described in just two patients, one with posteriorly rotated ears, and another with large low-set ears [20,21]. In patients with an unbalanced translocation between chromosome 13 and 18, a flat nasal bridge was commonly described. A long or broad nose was rarely seen [20,21,22,23,24]. Almost all the patients had dental problems and some of them had a high-arched palate [20,21,22,23]. Three of the five previously described patients had a short neck [20,21,24]. Some of them had a large sloping or prominent forehead [21,22]. All of these dysmorphic features are summarized in Table 2.

Our patient received thyroxine supplements for hypothyroidism, which was reported in only one of five patients in the available published papers [21]. We should also highlight that our patient is the first reported to have hypothyroidism at such a young age. All of these patients had ophthalmological manifestations, but our patient is the only one to have astigmatism and have this as her only ophthalmological manifestation at the time of investigation [7,20,21,22,23,24]. There is no evidence that iron deficiency is related to this condition, although feeding problems and low food intake in these patients can lead to nutritional deficiency. Anemia has not been reported in any other patient described in the literature. However, all the previously described patients had psychomotor development delay [20,21,22,23,24]. Despite the fact that our patient’s EEG showed nonspecific changes, she had no seizures, so there was no need for anticonvulsive medicine in contrast to three other patients described before [22,23,24]. The low levels of serum IgG antibodies for the patient’s age, which can be connected to 18p monosomy [7], require further involvement of a hematologist or immunologist. Also, the absolute number of memory B lymphocytes was marginally reduced compared to the reference range according to age, so this should be monitored in the future. One of the previously described patients had IgA deficiency, which is in accordance with the majority of patients with monosomy 18p who presented with immunodeficiency. On the other hand, there are no data about the immunological status of the other patients, just recurrent lower respiratory tract infections in one of the patients [7,20,22]. Like in almost every patient who has 18p monosomy [5,6], and every patient described who, has the unbalanced translocation between chromosomes 13 and 18, but not a deletion, the psychological testing of our patient suggests that the child’s psychomotor development was delayed by about 48 days [20,21,22,23,24]. This was mostly in terms of motor skills, but speech delay was present, which is in accordance with all of the other described patients [20,21,22,23,24]. Though a lower IQ was expected in our patient, as it was significantly reduced in four other reported cases, her age was a limiting factor in determining her IQ. Data on a patient’s IQ are missing in another paper [20,21,22,23,24]. Also, as skeletal disorders were described in three of the five previously described patients, it is recommended that our patient be closely monitored for that kind of anomaly in the future [20,21,24].

G-banding, a fundamental technique in cytogenetics, provides a clear visualization of the entire karyotype and can detect large-scale chromosomal changes. However, it lacks the resolution to detect small genetic changes (greater than 5–10 Mb). Moreover, high-quality metaphase spreads are necessary, which can be challenging to obtain from some cell types [25]. The most significant disadvantages of microarrays include the large number of probe designs based on sequences of low specificity, as well as the lack of control over the pool of analyzed transcripts [26]. Moreover, due to the targeted nature of microarrays, microarray testing relies on pre-defined probes to detect known genetic variations. Through a deeper exploration of the genes lost in the deletion and their possible roles in the observed phenotype, we can connect the genotype to the phenotype more explicitly. An absence of specific functional genes in the most distal part of 18p causes clinical manifestations of 18p- syndrome such as a round face and short stature. According to the Decipher base, out of a total of 191 genes, 16 genes are disease-associated genes (OMIM morbid genes). AFG3L2 is a candidate gene for hereditary spastic paraplegias or neurodegenerative disorders as well as spastic ataxia–neuropathy syndrome. Mutations in SMCHD1 are causative for the development of facioscapulohumeral muscular dystrophy type 2. Moreover, the PIEZO2 protein has a role in rapidly adapting mechanically activated currents in somatosensory neurons. Its abundant expression was detected in dorsal root ganglia sensory neurons, suggesting a potential role in somatosensory mechanotransduction. Thus, it is possible that the central hypotonia and assumed frog-leg posture lying down seen in our case might have resulted from the deletion of several genes such as AFG3L2, SMCHD1, and PIEZO2. In addition, the MC2R gene plays a role in immune function and glucose metabolism, so we assume that it could be related to our patient’s immune profile. For other genes located in the deleted 18p11.32 to p11.21 region (APCDD1, EPB41L3, GNAL, LAMA1, LPIN2, NDUFV2, PSMG2, TGIF1, THOC1, TUBB6, TYMS, and USP14), we could not find a direct correlation with the phenotype. Cytogenetic testing provides us with an opportunity to distinguish 18p- syndrome associated with unbalanced translocation (13;18) from other neurodevelopmental disorders featuring mild intellectual disability and short stature. Since there is no causative treatment for 18p monosomy, a timely diagnosis is crucial for the application of an appropriate treatment in order to alleviate specific symptoms and clinical manifestations. Moreover, parents can be counseled about their expectations of a certain syndrome, and, with them, review the genetic risks. Therefore, a multidisciplinary approach is necessary to identify the expected clinical features and increase our knowledge of 18p- syndrome, all in the aim of improving the patient’s quality of life.

A genetic counselor should predict the eventual clinical manifestations of the disease according to the patient’s genotype. A multidisciplinary approach is paramount and therapy must be individual. In addition, patients require lifelong monitoring [4,22]. As in our case, psychomotor stimulation is essential and could relieve symptoms and slow down the progression of disease, so a speech therapist, oligophrenologist, and physiatrist should be involved in the patient’s therapy from the early stages of their disease. Neurological examination and monitoring is necessary because of the different neurological conditions that these patients may have. Holoprosencephaly is a severe brain malformation, linked with poor prognosis and limited therapy options [4,9,10]. Patients with seizures require EEG monitoring and anticonvulsive therapy [22,23,24]. The early diagnosis of endocrine disorders like pituitary or thyroid dysfunction can help, especially at an early age, because the hormonal status has a great importance in metabolism and nervous system development and function. Feeding problems and a low food intake sometimes cause nutrient deficiencies or malnutrition; a pediatrician, gastroenterologist, or nutritionist should prescribe adequate supplementation and a diet plan. Refractive errors like myopia and hyperopia, astigmatism, strabismus, and other ophthalmological conditions sometimes need correction and timely interval ophthalmology check-ups. Also, cooperation with an immunologist is important, especially when common infections or autoimmune diseases are reported. Cardiac malformations are uncommon in 18p deletion syndrome, but require special attention and sometimes even surgical treatment [14,15]. Specific orthopedic or other surgical procedures depend on the severity and location of the anatomical abnormalities [22]. Psychological support for the patients and their families can help with dealing with the diagnosis and also coping with effects of treatment.

The prognosis of these patients is unpredictable and depends on their genotype–phenotype correlation [7]. An early diagnosis, an individual approach to every patient, and the timely treatment of specific manifestations can slow down the progression of the disease and improve the patient’s quality of life. For those patients with severe brain malformations, the prognosis is poor; otherwise, survival up to the sixth decade has been reported [4,21].

## 4. Conclusions

The present case report demonstrates the phenotypic spectrum of a rare variant of monosomy 18 caused by an unbalanced whole-arm translocation between chromosomes 13 and 18. Having in mind that this syndrome can be easily overlooked in a clinical setting, our study emphases the significance of cytogenetic testing to diagnose this disease, which has been described only five times in the literature.

## Figures and Tables

**Figure 1 diagnostics-15-00358-f001:**
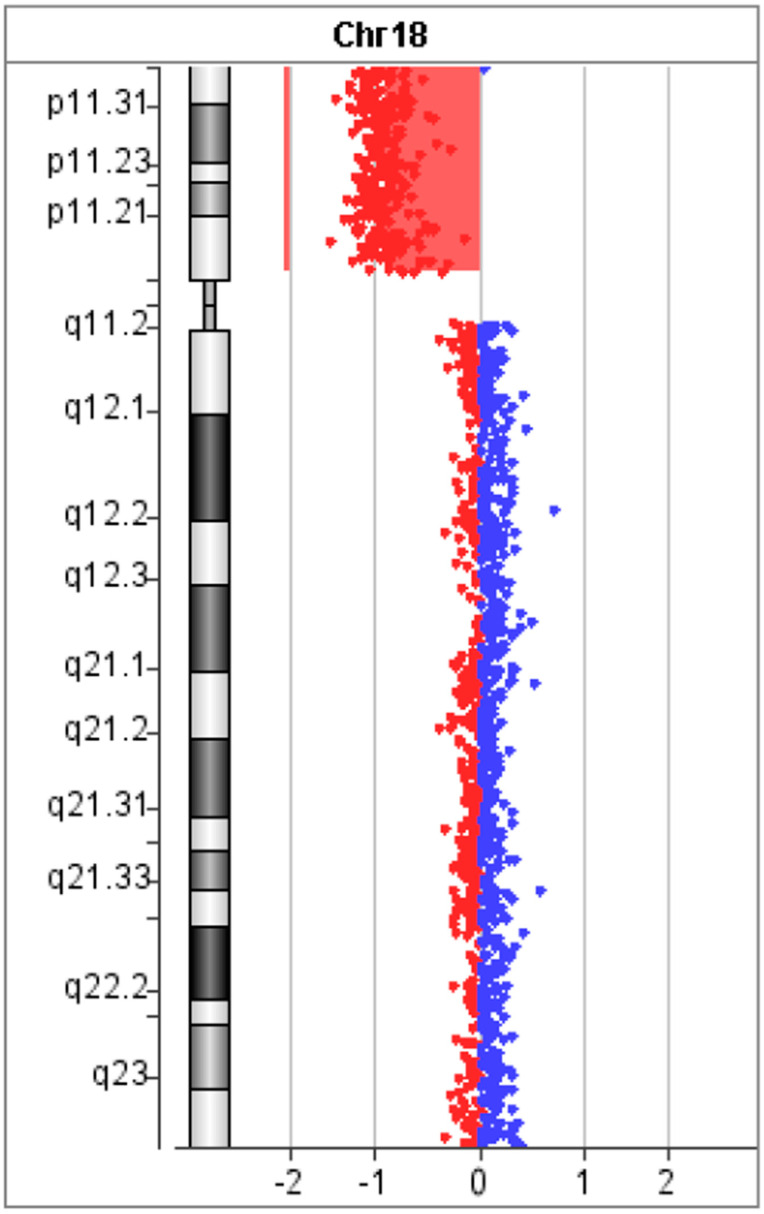
The result of array-CGH of the patient. Zero value indicates equal fluorescence intensity ratio between the sample and reference. Copy number losses shifted the ratio toward left (red) whereas copy number gains towards the right side (blue). Our result showed the deletion of short arm from 18p11.32 to p11.21 region (14.79 Mb).

**Table 1 diagnostics-15-00358-t001:** The analysis performed on the population of B lymphocytes determined by the electronic limiter of the flow cytometer based on the lateral scattering of the light beam and the intensity of expression of the CD19 molecule.

Subpopulations of B Lymphocytes (%)		Reference Values for Age 18 Months–4 Years
Total memory B lymphocytes (CD19+CD27+)	3.9	7.0–24.3
Naïve B lymphocytes (CD27−IgD+)	95.0	69.2–90.5
Double-negative B lymphocytes (CD27−IgD−)	1.1	1.2–8.3
Memory B lymphocytes without isotype switching (CD27+IgD+)	2.6	4.6–16.3
Isotype-switched memory B lymphocytes (CD27+IgD−)	1.3	2.7–12.5
**Subpopulations of B lymphocytes (/μL)**		**Reference values for age 18 months–4 years**
Total memory B lymphocytes (CD19+CD27+)	41	45–175
Naïve B lymphocytes (CD27−IgD+)	1007	212–1027
Double-negative B lymphocytes (CD27−IgD−)	12	10–56
Memory B lymphocytes without isotype switching (CD27+IgD+)	28	23–113
Isotype-switched memory B lymphocytes (CD27+IgD−)	14	20–93

**Table 2 diagnostics-15-00358-t002:** Facial dysmorphism of patients with monosomy 18p and unbalanced translocation between chromosome 13 and 18 based on their karyotype.

Cases	Case 1	Case 2	Case 3	Case 4	Case 5	Our Patient
s	45, XY, −13, 18, +t(13;18) (13qter→cen→18qter)	45, XY, der(13;18) (q10;q10)	45, XX, t(13p;18p)	45, XY, der(13;18) (q10;q10)	45, XY, der(13;18) (q10;q10)	45XX, t(13;18) (q12:p11.2)
Head shape	Normal circumference of head	Triangular broad face	Dolichocephaly	Brachycephaly	Round face	Microcephaly
Eyes	Strabismus	Strabismus, ptosis, epicanthic folds	Strabismus, hypertelorism	Ptosis, hypertelorism, protruding eyes	Strabismus, ptosis, hypertelorism	Almond-shaped upslanting palpebral fissures, epicanthic folds
Ears	Posteriorly rotated	Large low-set	NA	NA	NA	Normal
Nose	NA	Long and broad	Flat nasal bridge	NA	Flat and long	Depressed and broad nasal bridge
Mouth	Dental caries	Disarrayed teeth, dental caries	High arched palate	Hypodontia	NA	Deep and long philtrum, high-arched palate, delayed dentition
Neck	Short and broad	Short	NA	NA	Short	Normal
Other	NA	Large sloping forehead, alopecia	Prominent forehead	NA	NA	Light blonde hair, light complexion, broad forehead, light hyperpigmentation on the nasal tip
References (year)	Moedjono SJ et al. [20] (1979)	de Ravel TJ et al. [21] (2005)	Nema et al. [22] (2016)	Safavi et al. [23] (2019)	Choi et al. [24] (2022)	

## Data Availability

All data are contained within the article.

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
