# Peer review of "Monosomy 18p with Unbalanced Translocation Between 13 and 18 Chromosomes: First Reported Case in Serbia"

_diagnostics, 2025, doi:10.3390/diagnostics15030358_

Round 1
Reviewer 1 Report
Comments and Suggestions for Authors
Manuscript ID diagnostics-3358782, Case Report, Title
Monosomy 18p with unbalanced translocation between 13 and 18 chromosomes: a first case from Serbia
is a very interesting and unique case report of a very rare monosomy 18p syndrome, which has enormous significance for the correlation of phenotype and karyotype in the field of medical genetics. The manuscript had to be slightly modified before it was accepted for publication.
line 24 and 46- – mental retardation - change to intellectual dysfunction/disability
line 28 – shorten this sentence and emphasise only - astigmatism, which in this case is a unique clinical presentation
line 39 - start the sentence with a word and not a number
line 119 – 114 – this subtitles are not necessary, reviewer suggest to have just one subtitle Molecular cytogenetics with description of conventional and molecular karyotype (add molecular karyotype also) , differential diagnosis could be mentioned before molecular cytogenetic methods and results
line 123 –add section molecular cytogenetics
line 133 - insert in the section on molecular cytogenetics
add an figure of the chromosomes from
line 141 – add in the same paragraph of molecular cytogenetics
line 146 – please explain how it is possible that a deletion of almost 15Mb was not detected during karyotyping ?????
line 203 – error in the text
References should be described as follows, depending on the type of work:
- 1. Author 1, A.B.; Author 2, C.D. Title of the article. Abbreviated Journal Name Year, Volume, page range.
Reviewer 2 Report
Comments and Suggestions for Authors
This study presents a case report on a one-year-old female with partial monosomy of chromosome 18. The manuscript provides a detailed description of clinical observations and diagnostic findings, which are well-documented and thoroughly explained. I have a few minor comments for consideration.
1. Inclusion of Standard Karyotyping Results:: Cytogenetic analysis was performed as part of the study, but the results of the standard karyotyping are not visually presented. Including a figure of the karyotyping results as a main figure would enhance the clarity and completeness of the study.
2. Revisions to Molecular Karyotyping Figure: While the molecular karyotyping results are presented in Figure 1, the current figure is difficult to interpret due to low resolution and unclear text. I suggest replacing it with a higher-resolution image and moving it to the Supplementary Figures. In its place, a schematic diagram highlighting the chromosomal abnormality would more effectively convey the key findings to readers.
The comparison of findings from this case with other studies involving 18p deletion, as shown in Table 2, is particularly valuable for contextualizing the results. Overall, the results and discussion sections are well-structured and presented in a clear and comprehensible manner.
Reviewer 3 Report
Comments and Suggestions for Authors
Here is a critique of the article, "Monosomy 18p with Unbalanced Translocation Between 13 and 18 Chromosomes: A First Reported Case from Serbia":
1. As a single-case report, the findings cannot be generalized. While this is inherent to case reports, the authors could have provided a broader context by discussing additional cases in more depth.
2. Although comparisons to previously reported cases are insightful, the paper could have analyzed the current case in more depth, particularly by exploring potential mechanisms underlying the phenotypic variability.
3. The discussion on treatment is brief, stating there is no causative therapy for Monosomy 18p without delving into the specifics of managing associated complications like hypothyroidism, anemia, and developmental delays. Practical insights into long-term care would have enhanced the paper's clinical utility.
4. The article identifies a 14.79 Mb deletion but does not explore in detail the functional implications of the affected genes. This represents a missed opportunity to connect genotype to phenotype more explicitly. Provide a deeper exploration of the genes involved in the deletion and their possible roles in the observed phenotype.
5. While the diagnostic approach is detailed, there is little critical evaluation of the limitations of the cytogenetic techniques used (e.g., resolution constraints of G-banding or limitations of the microarray approach).
6. Although the need for a multidisciplinary approach is mentioned, there is no elaboration on how specialists like hematologists, ophthalmologists, or genetic counselors could collaborate effectively for this condition.
7. While the case details the initial findings and interventions, there is little discussion on the long-term prognosis or follow-up, leaving the reader without a sense of how the patient’s condition evolved. Elaborate on the contributions of various specialists in the care of patients with Monosomy 18p. Include more detailed comparisons with other cases of 18p- syndrome, particularly focusing on differences in phenotypic expression.
8. Several sentences lack clarity or contain grammatical errors (e.g., “There is no evidence that iron deficiency can be related with this condition, except for low food intake and feeding problems”). The formatting of the references is inconsistent, and some citations lack sufficient detail for easy verification.
Round 2
Reviewer 3 Report
Comments and Suggestions for Authors
Accept in present form